# Competitive co-diffusion as a route to enhanced step coverage in chemical vapor deposition

Arun Haridas Choolakkal ®, Pentti Niiranen ®, Samira Dorri, Jens Birch ® & Henrik Pedersen ® ✉

Semiconductor devices are constructed from stacks of materials with different electrical properties, making deposition of thin layers central in producing semiconductor chips. The shrinking of electronics has resulted in complex device architectures which require deposition into holes and recessed features. A key parameter for such deposition is the step coverage (SC), which is the ratio of the thickness of material at the bottom and at the top. Here, we show that adding a co-flow of a heavy inert gas affords a higher SC for deposition by chemical vapor deposition (CVD). By adding a co-flow of Xe to a CVD process for boron carbide using a single source precursor with a lower molecular mass than the atomic mass of Xe, the SC increased from 0.71 to 0.97 in a 10:1 aspect ratio feature. The concept was further validated by a longer deposition depth in lateral high aspect ratio structures. We suggest that competitive co-diffusion is a general route to conformal CVD.

The architecture of modern semiconductor devices requires deposition on, and even filling of, topologically very complex structures with films of various materials with different properties[1–4] The challenge of depositing these films increases with the aspect ratio (AR), i.e., the ratio of depth to width, as well as with smaller scale of the features. The success of the film deposition is typically measured by the conformality of the film, which in turn is quantified by the step coverage (SC), defined as the film thickness at the bottom of the feature divided by the film thickness at the top of the feature[5]. For perfect conformality, the film is as thick at the bottom as it is at the top, i.e., SC = 1. The best approach to deposit conformal films is often by atomic layer deposition (ALD). Since ALD uses self-terminating surface chemical reactions[6,7], an ALD process will be highly conformal if given enough time for the surface reactions to saturate all surfaces[8,9]. However, ALD processes are not available for all materials. Carbides is one example of materials systems where only a few ALD processes have been reported due to the lack of carbon precursors rendering self-terminating surface chemistry[10,11]. If the surface chemistry is not self-terminating, the deposition is instead referred to as chemical vapor deposition (CVD). Most CVD processes afford sub-conformal films with SC < 1 due to the less controlled surface chemistry[12]. Very effective strategies for controlling the gas transport and reaction probabilities inside high AR features have been demonstrated, allowing CVD to not only deposit conformal films (SC = 1), but also superconformal films (SC > 1)[13–15].

The deposition rate in CVD depends on the flux of film forming species and energy supplied to the surface[16,17]. In high AR recessed features, the challenge is to maintain a uniform flux over the surface with supply of film forming species from the opening of the features. Inside the features, the partial pressure of film forming species decreases the farther away from the nominal opening, rendering a higher incident flux with a higher deposition rate at the feature opening which leads to SC < 1. The reaction probability can be lowered by using lower temperatures, using precursors with low sticking coefficients[18] or by adding a surface inhibitor[10,19,20]. These two methods will promote gas phase diffusion down the feature and increase the SC. In CVD processes using two precursors, partial pressure tuning of precursor gases can also improve the SC in high AR features[14], albeit with the risk of affecting the compositional homogeneity of the film.

Instead of trying to alter the reaction probabilities for surface chemical reactions, we seek to promote the diffusion of precursor molecules down the recessed features to allow for conformal deposition also at higher temperatures. In this quest, we considered the

Department of Physics, Chemistry and Biology, Linköping University, Linköping, Sweden. ✉e-mail: henrik.pedersen@liu.se

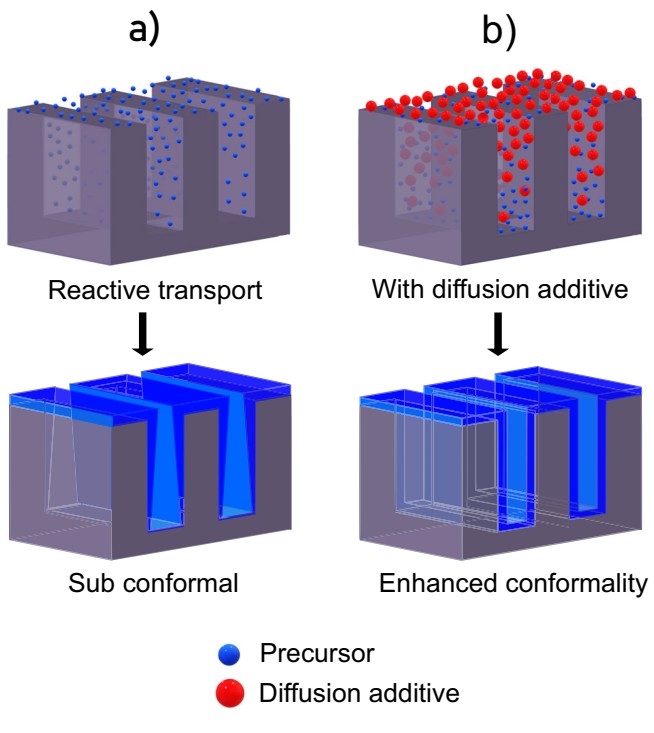

**Fig. 1 | Schematic illustration of the competitive co-diffusion model.** Diffusive transport of precursor in high aspect ratio features and the resulting step coverage (**a**) without diffusion additive and **b** with a diffusion additive. It illustrates the hypothesized diffusion mechanism that lighter molecules diffuse faster than heavier molecules in binary gas mixture at local thermodynamic equilibrium. The hydrogen gas that sets the background pressure is not depicted in the schematic.

kinetic molecular theory which assumes that the temperature of a system in the absolute scale is proportional to the average kinetic energy of its particles. A local thermodynamic equilibrium can be assumed at, and above the substrate surface, in thermal CVD processes[21]. The gaseous species above the surface will then all have the same temperature, and hence the same average kinetic energy. For a binary mixture of two gases, A and B, this can be expressed as:

$$\frac{1}{2}M_A V_A{}^2 = \frac{1}{2}M_B V_B{}^2 \qquad (1)$$

Where $V_A$ and $V_B$ are the average molecular velocity of gas A and B, respectively, and $M_A$ and $M_B$ are the molecular mass of gas A and gas B, respectively. Equation (1) can be rearranged to Grahams law of diffusion[14,22,23]:

$$\frac{\text{Diffusion rate of gas A}}{\text{Diffusion rate of gas B}} = \sqrt{\frac{\text{Molecular mass of gas B}}{\text{Molecular mass of gas A}}} \qquad (2)$$

since the diffusion rate and the partial pressure of a given molecule control the flux of that molecule to the surface. If that molecule is a CVD precursor, the diffusion rate and the partial pressure also controls film deposition rate[17]. From these consideration, we seek to explore if this diffusion phenomenon can be used to improve film uniformity and SC of CVD in recessed features, without reducing reaction probabilities by low temperature or surface inhibitors. We speculate that a molecule with higher mass than the film precursors, and that is inert to the deposition chemistry, could be used as a diffusion additive (Fig. 1) for this approach. A requirement is that it does not introduce impurities in the film.

We have recently shown that conformal, i.e., SC = 1, amorphous boron carbide films can be deposited in 8:1 aspect ratio trenches from triethylboron, $B(C_2H_5)_3$ (TEB), in a hydrogen ambient at 450 °C substrate temperature and 5 kPa total pressure[24]. We noted 450 °C as the upper temperature limit for SC = 1, and that the SC decreased with increasing temperature. On the other hand, film density improved at temperatures above 450 °C, making higher temperature deposition desirable. The molecular mass of TEB is 98 amu, and its gas phase chemistry renders only lighter species[25]. Thus, xenon (Xe), with an atomic mass of 131 amu, is a suitable diffusion additive to test our above hypothesis also at higher temperatures. The process pressure was chosen to be in the transitional flow regime, allowing to find a balance between effective mass transport and desired gas phase collisions between precursor molecules and the diffusion additive.

## Results

Fig. 2 shows the comparison of SC between samples deposited for 60 min in three different deposition conditions, i.e., without Xe, with 100 sccm co-flow of Xe and with 100 sccm co-flow of Ar. We maintained the total reactor pressure for all the experiments, allowing us to operate the processes at the same Knudsen number, facilitating a meaningful comparison. It can be observed from the scanning electron microscopy (SEM) measurements that the SC was improved from 0.71 to 0.97 when Xe co-flow was added. A control experiment with a co-flow of Ar, with atomic mass of 40 amu, i.e., lower than the TEB molecule, afforded a SC of 0.69, i.e., very similar result as without any co-flow of inert gas. This strongly suggests that it is not the inertness of Xe that enhanced the SC, but rather the atomic weight, in accordance with the hypothesis of competitive diffusion. We speculate that a reduction of the surface residence time and enhanced desorption rate for reactive intermediates caused by surface interactions with the heavy Xe atoms can also contribute to the observed results.

Another way to interpret the results shown in Fig. 2 is through the Langmuir model[26,27] of mass balance at the surface sites. This model sees through the perspective of local precursor concentration and the consequent impact on precursor molecule occupancy at surface sites for the deposition process. With Xe co-flow, the differential diffusion rates of TEB and Xe result in a concentration gradient along the trench depth, as shown in Supplementary Fig. 4. This, in turn, modifies the local dilution of the TEB molecules at the surface site along the trench depth in addition to the modification in the absolute local influx. At the trench top, a minor thinning of the film, i.e., from 453 nm to 436 nm, suggests a reduced Langmuir adsorption of TEB due to the presence of Xe. Conversely, at the trench bottom, an enhanced concentration of TEB enhances the Langmuir adsorption rate, facilitating more precursor molecules to attach and promote film growth. Furthermore, decrease in film thickness from 453 nm to 442 nm with 50 sccm Xe flow (see Supplementary note 1) aligns with this observation. A similar observation can be made with Ar (Fig. 2). However, unlike Xe, Ar does not cause any differential dilution to result an enhancement in the SC. On the other hand, the probability of lighter Ar undergoing collisions with TEB is much less than that of collisions with Ar itself due to the two orders of magnitude higher partial pressure for Ar, preventing possible diminution of the SC. Whereas, in the case of Xe and TEB, the dynamics are reversed, the lighter TEB molecules are benefited from the elastic collisions with the excess heavier Xe with a higher probability than undergoing collisions with TEB molecule itself. The overall pathway that resulting in the observed SC enhancement is likely a result of collective effect of more than one phenomenon.

The films were confirmed to be boron carbide by X-ray photoelectron spectroscopy (XPS), see Supplementary note 2. The elemental composition, by time-of-flight elastic recoil detection analysis (ToF-ERDA), of films deposited in hydrogen ambient at 550 °C with a Xe co-flow is approximately 82 at.% B, 17 at.% C, 0.5 at.% O, and 0.2 at.% H, giving a stoichiometry of about $B_{4.7}C$. No Xe could be detected in the films. A small, but notable, 1 at. % increase in carbon content was noticed for the films deposited with the Xe co-flow, see Supplementary

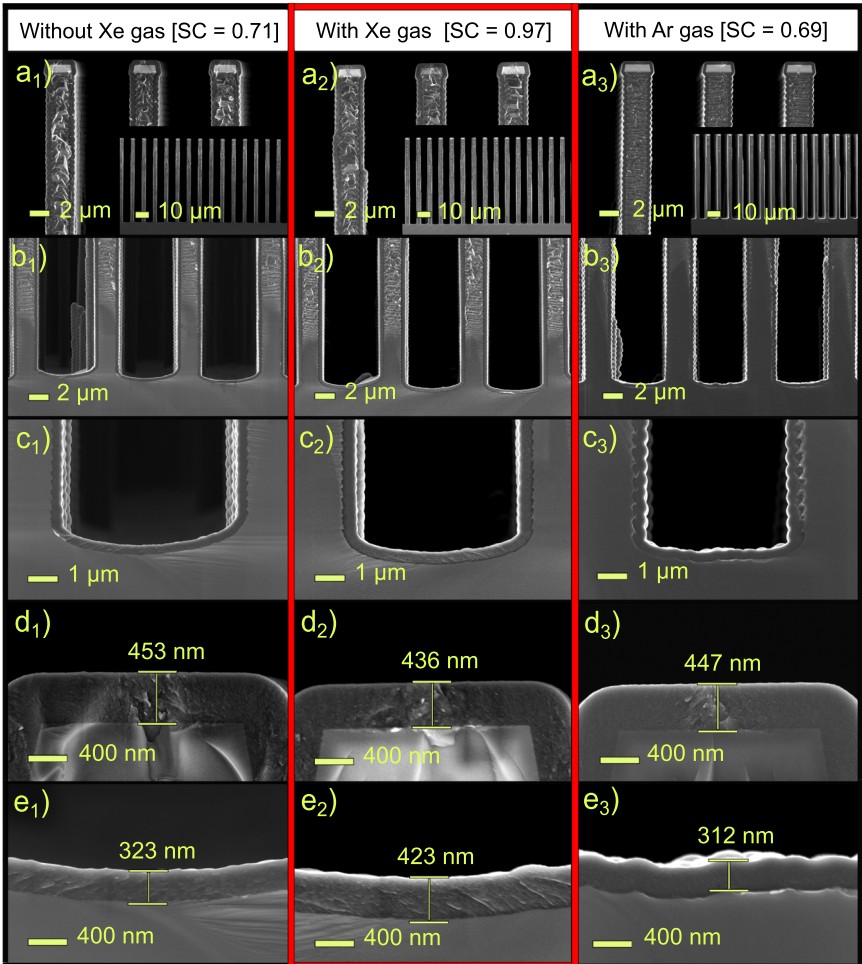

**Fig. 2 | Electron micrographs showing film thickness in trenches.** SEM micrographs show film deposited in 10:1 aspect ratio features at 550 °C substrate temperature for 60 min, without Xe gas ($a_1$–$e_1$), with Xe gas ($a_2$–$e_2$), and with Ar gas ($a_3$–$e_3$). In (**a**) 10:1 aspect ratio trench pattern in Si substrate and top of the Si pillar, **b** bottom of the Si pillar, **c** magnified view of the bottom surface, **d** film thickness at the top surface, and **e** film thickness at the bottom surface. The red border is given as a guide for eye to compare step coverage (SC) of the films deposited with and without Xe gas.

note 2. We attribute this to a small change in the ratio of β-hydride elimination to hydrogen assisted ligand elimination, which are the two different decomposition pathways for TEB[25]. The β-hydride elimination forms more reactive ethylene ($C_2H_4$) which could contribute to higher C in the film, while the hydrogen assisted elimination forms less reactive ethane ($C_2H_6$)[25]. The addition of Xe induces a small reduction in the hydrogen partial pressure, 4887 Pa $H_2$ and 113 Pa Xe, as compared to 4999 Pa $H_2$ without Xe co-flow, which will affect the gas phase chemistry of TEB. In our previous work, we observed that the boron-to-carbon ratio shifted from $B_4C$ to $B_3C$ when transitioning from purely Hydrogen to purely Argon ambient[24].

In addition to higher SC with a Xe co-flow, we also note a more uniform film deposition over a flat substrate surface, as shown by the deposition rate mapped over 10 mm × 100 mm area for films deposited with and without Xe co-flow (Fig. 3). The deposition rate distribution ranged from 7.6 nm/min to 0 nm/min without Xe, while the introduction of a Xe co-flow yielded a more uniform film deposition with a deposition rate ranging from 7.3 nm/min to 1.5 nm/min. These results suggest that the Xe co-flow enhances the precursor diffusion also on a lateral macroscale in the CVD reactor, not only on a vertical microscale in the trenches. We attribute the changes in growth rate distribution over the large area to the enhanced diffusion on a reactor scale. Additionally, this could also be due to a reduction in surface residence time for reactive intermediates, caused by the addition of heavy Xe gas, which lead to a higher lateral spread. This implies that

reactive species spend less time absorbed on the surface, allowing for a more effective lateral spread of the gas mixture across the substrate. Alternatively, since solid-gas interface also influence the temperature uniformity in the reactor[28], the modified boundary layer induced by the more viscous Xe flow could potentially favor more uniform film deposition, even though this was not an intentional objective. These kinetic and thermodynamic factors could collectively contribute to the observed phenomena, indicating a potential interplay of mechanisms that affect precursor diffusion both vertically and laterally within the CVD reactor.

The as grown films were X-ray amorphous. X-ray reflectivity (XRR) measurements show that the samples have similar densities of $2.25 \pm 0.2\%$ g/cm³ with an interface roughness of less than 0.9 nm whether deposited with or without Xe addition. The density value observed is comparable to the bulk density value of 2.48 g/cm³ reported for boron-rich boron carbide[29,30]. Residual film stress obtained by the wafer curvature method[31–33] shows a tensile stress of ~0.8 GPa and ~0.3 GPa for the films deposited with and without Xe, respectively. Although the residual stress exhibits a relatively large increase upon growth with Xe, it is noted that the absolute stress level of ~0.8 GPa is still low, particularly for boron carbide films[34]. An increased tensile film stress when adding Xe to the process can be attributed to the associated increase in partial pressure of the reactive species which, in turn, leads to a shorter diffusion length and reduced admolecular mobility during film nucleation[35–37]. Such conditions

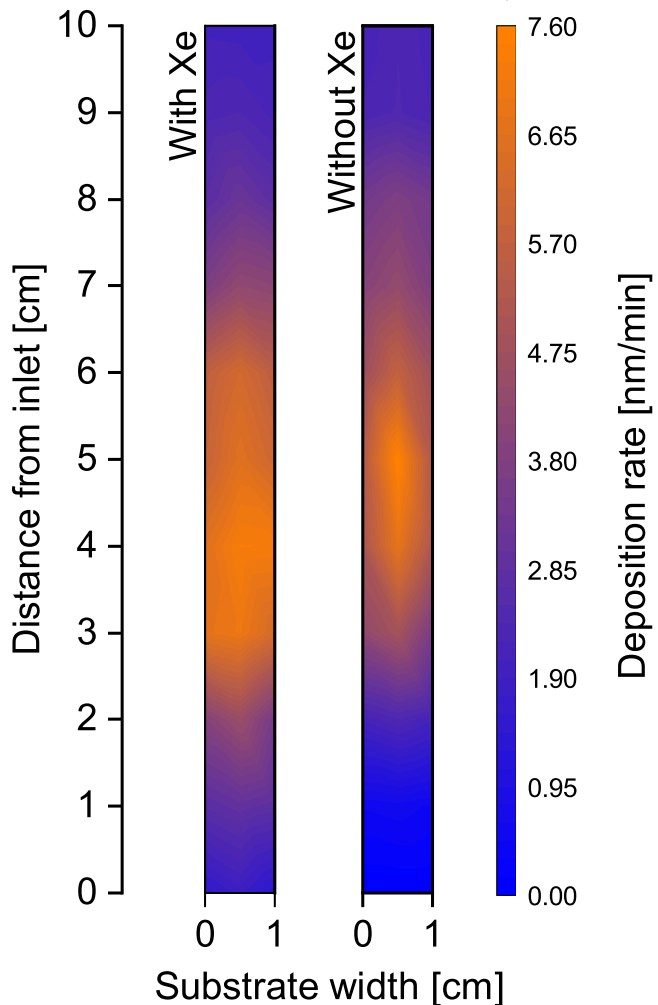

**Fig. 3 | Film uniformity over the sampleholder area in the CVD reactor.**
Deposition rate mapping obtained for the B$_{4.7}$C films deposited on 10 mm × 100 mm silicon substrate with and without Xe at 550 °C substrate temperature. It shows increased area of uniform film deposition in the reactor with the addition of Xe. Source data are provided as a Source Data file.

as 50:1. The CVD process demonstrated reduced penetration depths in the structures with extended lengths. This phenomenon is likely attributable to the inhibited back diffusion of precursor, suggesting the considerable length of these structures, though a deeper understanding requires further studies. Consequently, it is possible to infer that conformal growth might be attainable even in structures with aspect ratios surpassing 50:1.

The lateral high aspect ratio structures allowed the top membrane to be lifted off, enabling film characterization by XPS also inside the structure. The B 1$s$, C 1$s$ and Si 2$p$ XPS core level spectra from within the structure are presented in Fig. 5, showing very similar peak positions for the signals probed from the film deposited both outside and inside of the lateral high aspect ratio structure. The intensity of the Si 2$p$ peak increases inside the structure which is explained by the presence of uncoated Si pillar tops that come in the XPS beam spot area. Since the top membrane was removed after the deposition, the pillar tops are free of boron carbide deposition. Although the B 1$s$ and C 1$s$ core level spectral intensity decreases inside the structure due to the higher XPS yield of Si, due to the low film thickness, the peaks are at the same positions and without any chemical shift.

## Discussion

We demonstrate a competitive co-diffusion CVD concept for conformal film deposition by using Xe as an inert diffusion additive in CVD of boron carbide from the single source precursor TEB. Our results show an increase in the step coverage from 0.71 to 0.97 in 10:1 aspect ratio feature when Xe was added to the process. At present, the exact mechanism for this increase in SC is not understood. We suggest that possible mechanisms could be that the heavier Xe atoms affects the diffusion of the lighter precursor molecules by competitive co-diffusion, and that the heavy Xe atoms reduces the surface residence time by enhancing desorption rate for reactive intermediates by surface collisions. Further experimental and modeling studies are needed to fully determine the mechanism, which could also be a combination of at least these. The elemental composition, chemical environments, and density of the films were not significantly affected by the addition of Xe gas. The concept was further demonstrated by deposition in lateral high aspect ratio structures where addition of Xe afforded film deposition corresponded to conformal film in at least a 50:1 aspect ratio feature. We foresee that this approach can improve conformality in many CVD processes.

## Methods

The thermal CVD process was conducted in a horizontal hot-wall CVD reactor. TEB (semiconductor grade, from SAFC Hitech) was used as single source precursor for B$_x$C and palladium membrane purified H$_2$ as a carrier gas and co-reactant. The TEB was kept in a stainless-steel bubbler, in a thermostat bath at 0 °C to maintain a vapor pressure of ~1.65 kPa for a stable precursor delivery achieved by bubbling H$_2$ through the TEB liquid. A co-flow of Xe (99.998%) was introduced as diffusion additive. Ar (99.997%) was used in control experiments. Some experiments were conducted by delivering Xe in a pulsed manner, see Supplementary note 3. This was done by pulsing Xe gas for 1 s after every 30 s.

The films were deposited both on polished Si (100) and Si substrates with a 10:1 aspect ratio trench pattern, ~6 µm wide and ~60 µm deep. The $10 \times 10 \times 0.5\,mm^3$ and $100 \times 10 \times 0.5\,mm^3$ polished Si substrate pieces and $10 \times 10 \times 0.5\,mm^3$ Si substrates, patterned with 10:1 AR features were all cleaned using an ultrasonic bath, 3 min each in acetone and ethanol, and finally blow-dried with nitrogen gas. The substrates were loaded into the SiC coated graphite susceptor, which had a maximum substrate area of $30 \times 100$ mm. Then the reactor was pumped down, backfilled with H$_2$, and heated to deposition temperatures with a 2000 sccm flow of H$_2$ at 5 kPa regulated by a throttle valve on the process pump. The temperature was stabilized for 5 min at the

typically lead to the formation of less dense films and development of more tensile growth stresses.

The effect of the Xe co-flow was further studied by depositing in lateral high aspect ratio structures with 500 nm gap height, schematically shown in Fig. 4a. The analysis of the top view SEM depicted in Fig. 4b illustrates the penetration depth within the lateral high aspect ratio (LHAR) structure with 100 µm lateral depth, both with and without the addition of Xe co-flow. The micrographs show that the penetration depth reaches approximately 15 µm in the absence of Xe, but with the introduction of Xe co-flow, this depth extends to about 25 µm. Complementing this, the energy dispersive X-ray (EDX) mapping presented in Fig. 4c, along with the top view SEM micrographs of the mapped area, corroborates the film deposition within the structure. The presence of B and C traces that detected up to about 15 µm lateral depth was expanded to 25 µm when Xe co-flow was added. Furthermore, as shown in Fig. 4d, while the EDX line scan across the LHAR depth for the sample deposited without Xe shows a penetration depth up to 15 µm, a deposition depth of up to 25 µm depth was observed with Xe addition. Beyond this, a graded thickness was observed, ranging from 25 µm to approximately 35 µm.

These observations imply that the deposition process could achieve conformality within features possessing an aspect ratio as high

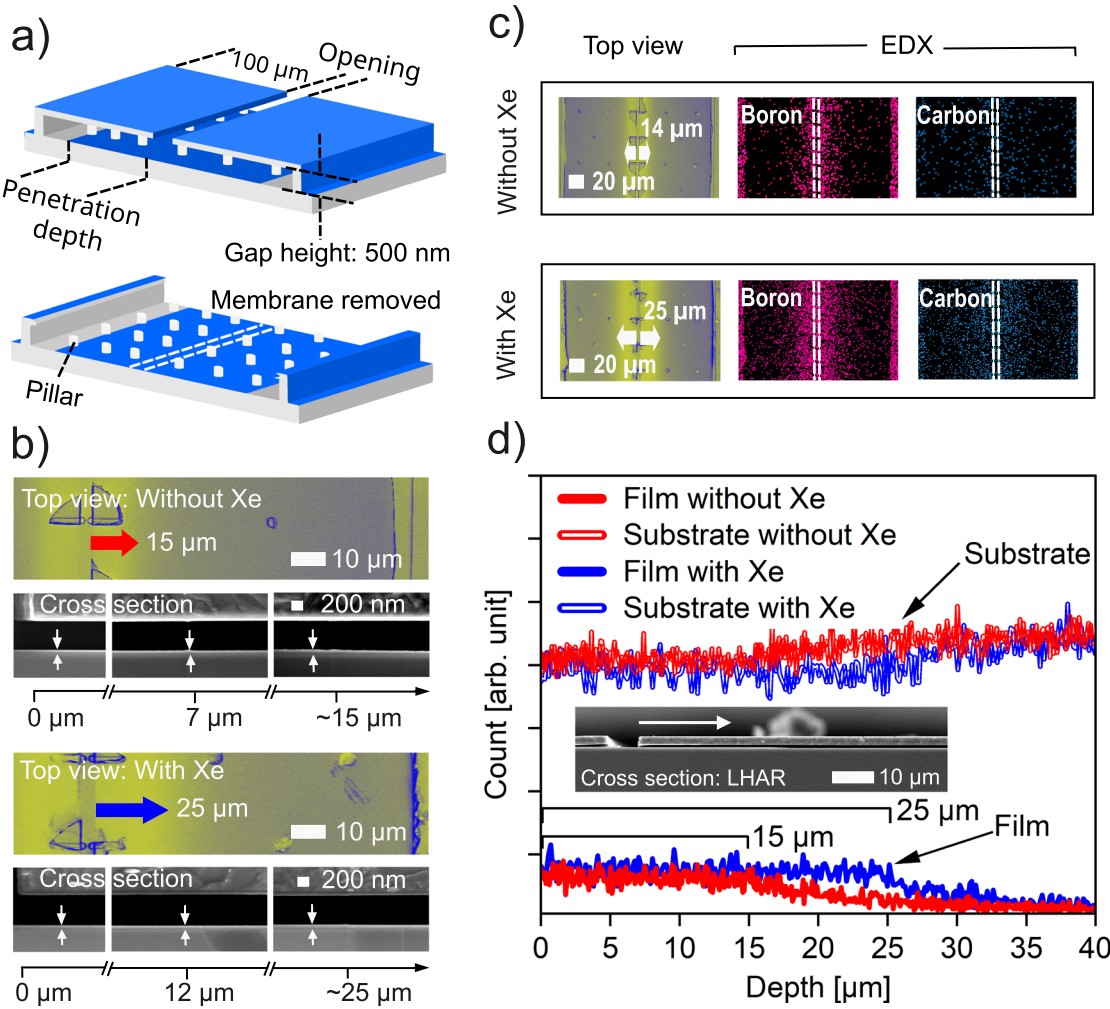

**Fig. 4 | Boron carbide deposited in LHAR structures. a** Schematic shows the details of the LHAR (lateral high aspect ratio) chip. **b** Contrast enhanced to duo-chrome top view SEM micrographs show penetration depth together with cross-sectional view. **c** EDX elemental maps showing boron and carbon distribution within the structure. **d** EDX line scan across the LHAR depth obtained for the samples deposited with and without Xe co-flow. Source data are provided as a Source Data file.

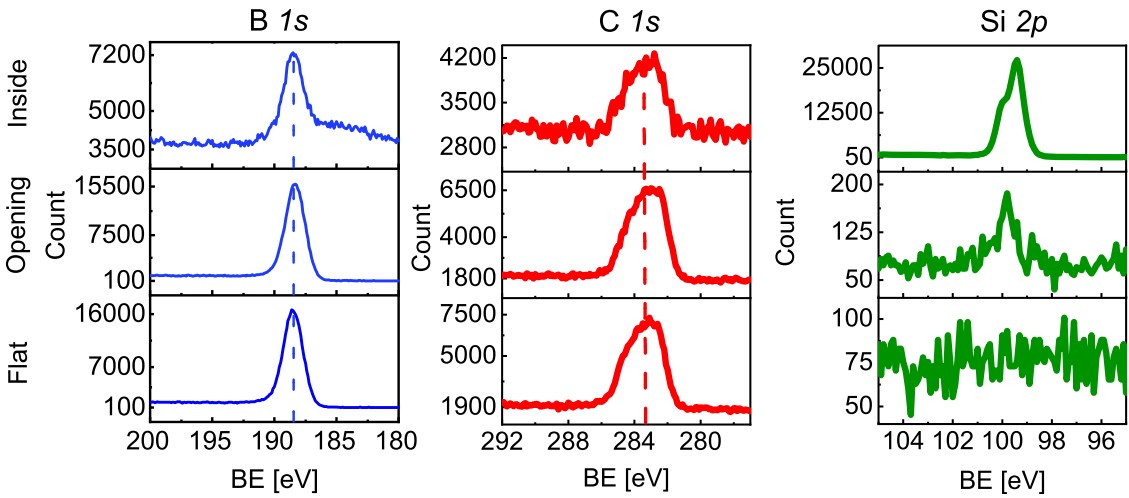

**Fig. 5 | Analysis of films deposited inside of the LHAR structures.** XPS spectra obtained for the film deposited inside the structure, at the structure opening, and, for comparison, on a flat substrate. The dashed vertical line shows the expected binding energy (BE) of the peak. Source data are provided as a Source Data file.

deposition temperature prior to the deposition process. Then a 1 sccm flow of TEB, to give a partial pressure of 1.18 Pa, was added together with 100 sccm of Xe or Ar, giving a noble gas partial pressure of 18 Pa, to start the deposition. The total pressure was kept at 5 kPa for all depositions, resulting in a transitional flow with a Knudsen number of about 1 corresponding to the 10:1 aspect ratio microstructure. The susceptor temperature was monitored by a pyrometer (Heitronics) which was connected in loop with the temperature controller. A few depositions were carried out using lateral high aspect ratio structures (Pillar-Hall[38] chips from Chipmetrics) as substrate. The top membranes of the chips were removed after the deposition using scotch-tape. The deposited films both inside and outside of the later high aspect ratio structure were characterized using XPS, EDX, and SEM. The chip was then broken perpendicular to the opening for cross-sectional analysis by SEM.

A scanning electron microscope (LEO 1550 Gemini) with 3 kV accelerating voltage and an in-lens secondary electron detector was used to micrograph and measure the film thickness over the 10:1 AR trench depth. The deposition rate mapping over 10 mm × 100 mm was accomplished by verifying the thickness through cross-section SEM measurements. The samples were broken into multiple pieces to perform cross-sectional thickness measurements with 1 cm intervals. Three measurements (left edge, center, and right edge) were performed at each cross-section. An energy dispersive x-ray detector with an atmospheric-thin-window in the same chamber was operated to obtain the elemental mapping and line scan over the pillar-hall structures. A 15 kV accelerating voltage was used for the EDX measurements. The elemental compositions of the samples deposited with and without Xe were measured using time-of-flight(ToF) ERDA. The measurements were carried out with 36 MeV Iodine ($^{127}I^{8+}$) ions as projectile beam and with a ToF detector. The extracted ToF-ERDA histograms were converted to elemental concentration-profiles using the Potku code[39]. Substrate curvatures were determined from rocking-curve measurements using a PANalytical EMPYREAN diffractometer with a Cu X-ray tube, operated at 45 kV and 40 mA. A high-resolution 4-bounce Ge (220) crystal monochromator was used on the primary side and an open detector was used on the secondary side. XPS was used to study the chemical environment in the films using an AXIS Ultra DLD, Kratos Analytical. The XPS analysis chamber was equipped with an Al $K_\alpha$ monochromatic radiation and an $Ar^+$ sputtering source. The X-rays were produced, using an anode current of 10 mA and an anode voltage of 15 kV. To remove surface contaminants, sputtering was employed, using an $Ar^+$ energy of 0.5 keV. The data obtained from the measurement was analyzed, using Casa XPS[40] software. Since the material was in an amorphous phase and amorphous carbon phase was present, the $sp^2$ hybridized C−C peak position was used to calibrate the XPS spectrum and was set to 284.5 eV[41]. The XPS chamber was used to carry out ultraviolet photoelectron spectroscopy[42] (UPS) measurements, which was done by operating a UV source (21.22 eV) in order to estimate the work function of the material. The film density was measured by XRR using Cu $K_\alpha$ radiation with the Cu $K_\beta$ filtered off by a nickel filter. An X'Pert MRD diffractometer with 1/32° divergence slit and a 0.27° plate collimator followed by a proportional detector were used to record the data. The measured data were fitted for thickness and density of the films using PANAlytical X'Pert Reflectivity software.

## Data availability

Source data are provided as a Source Data file: https://doi.org/10.6084/m9.figshare.26934448.

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

## Acknowledgements

Financial support by the Swedish research council under Contract No. 2018-05499, granted to J.B. and H.P. is gratefully acknowledged. Financial support from the Swedish Government Strategic Research Area in Materials Science on Advanced Functional Materials at Linköping University (Faculty Grant SFO-Mat-LiU No. 2009-00971), granted to J.B. and H.P., and support from the Swedish research council VR-RFI (No. 2019-00191) for the Accelerator based ion-technological center in Uppsala are gratefully acknowledged.

## Author contributions

A.H.C. initially conceptualized the idea of competitive co-diffusion and refined it together with H.P. A.H.C. and H.P. designed the experiments and A.H.C. did all the experiments. P.N. did the XPS measurements and S.D. assisted in film stress measurements. A.H.C. evaluated all characterization data and wrote the first draft of the manuscript. J.B. and H.P. refined the manuscript together with A.H.C.

## Funding

## Competing interests

A.H.C., J.B., and H.P. have filed a patent application based on the results presented in the manuscript. The remaining authors declare no competing interests.
