## [Transparent Peer Review file · Nature Communications]

Competitive co-diffusion as a route to enhanced step coverage in chemical vapor deposition

Corresponding Author: Professor Henrik Pedersen

Version 0:

Reviewer comments:

Reviewer #1

(Remarks to the Author)

The authors describe a method to increase step coverage/conformality in a boron carbide CVD process. This is a known issue and the solution of the authors is very original. The method consists of co-pulsing a heavy inert gas, here Xe, to increase the diffusion of the single-source precursor. This is an exciting idea, and I believe this could lead to novel and exciting research ideas and applications.

The paper is well-written and concise, and all claims are supported by measurements. I appreciated the use of LHAR structures as these are the state of the art characterization methodology for this type of problem. Below some comments and suggestions.

Major comments:

A cross-section of the LHAR structure shown in fig. 4 a/b/c is missing, but should be added. (maybe to the supporting information if space is an issue.) This would be very informative as it shows an edge case where the Xe-assisted process is "failing". I also strongly suggest an additional comparison of the obtained profiles/cross-sections with another LHAR substrate, but without Xe. That way the merits of the Xe addition can be fully appreciated.

In the supporting information, a diffusion model is developed and used to simulate the partial pressure. The authors state that surface reactions are ignored in this model. Is there any knowledge of TEB surface reaction coefficients that warrants such simplification? e.g. in the LHAR structure of 100 micron (AR 200) this simplification probably does not hold?

It seems the H₂ partial pressure has an effect on film composition through ligand elimination. What is the effect of the Xe co-flow on the H₂ diffusion? Authors should distinguish on the effect of Xe on both TEB and H₂ in their discussion. I feel this distinction is somewhat lacking now, both in the main text and supporting info.

Minor comments and suggestions:

frame b of figure 1 visually suggests that adding Xe leads to a higher pressure, but from the text I understood that the pressure remains the same. Soft suggestion to maybe reconsider this figure, so it includes H₂?

Fig 4 d/e could use some annotation, as the contrast film/substrate is not very high

Methods: "some experiments were conducted by delivering Xe in a pulsed manner". Which ones? Are these results displayed in the main text? Recommendation to clearly indicate which results had the Xe pulsed.

I believe equation S1 is used both for Xe and TEB diffusion. If this were stated more clearly in the text it would benefit the ease of reading (unless I misunderstood in which case the authors should mention why TEB is not included in their model).

Fig. S2: the markers showing the flux ratio are not very effective because they are too close to each other.

Reviewer #2

(Remarks to the Author)

This manuscript reports an interesting and unobvious result: in the deposition of boron carbide films by thermal CVD, the coating profile within a deep recessed structure (a trench) can be made more conformal by adding an inert gas (Xe) to the process gases. The experimental data are credible, but the generality of the result and how it may vary over the large area of the substrate are not reported.

The authors offer an explanation, which they call a hypothesis, in which (i) they state incorrect physics -- that gas phase diffusion may increase upon the addition of the Xe gas -- and (ii) they do not consider the many crucial steps leading to the observed result.

Including an incorrect and seriously deficient attempt at an explanation seriously weakens the credibility of this work. The experimental result alone, in the absence of a claimed explanation, is of some interest. Far better would be to advance a sound, well-articulated, and well-supported hypothesis.

A selection of specific concerns includes:

- In a microtrench, molecular collisions are typically with the wall (molecular flow conditions), not with other molecules in the gas phase. That can be estimated by simple mean free path calculations, which the authors have not done. Under molecular flow, the scattering effect of an inert gas is normally small and can be neglected.

- Outside of the trenches, the overall reaction kinetics are clearly varying upon addition of Xe, as evidenced by the significant change in growth rate over the large area of the substrate (Fig. 3a, "with Xe"). The reasons for this are not discussed.

The experimental details state that the plug-flow system is run at constant total pressure as controlled by a throttle valve with a feedback system. In this case, the pumping speed and therefore the partial pressure for each gas will vary with process conditions in a way that is not known unless non-trivial calibrations are performed.

I appreciate that it is not realistic to attempt to determine "all" the details of the reactive species mix. But there should be some discussion based on correct physics and chemistry, and preferably some parametric experiments, to sort out what is most important in this effect.

- I have inserted numerous other comments in the manuscript.

As written this work should be rejected. However, I have indicated required major revision in hopes that the authors can remedy the deficiencies.

In addition to what is written in the manuscript, below are other detailed concerns.

Some critical information is missing:

- What is the actual TEB flow/pressure? How does it compare to the inert gas flow and is H₂ in excess (the authors pointed out that TEB partially reacts with H₂)
- Growth rate: there is no absolute thickness mentioned in the text for the data on Fig.3, and the deposition time is not mentioned either.
- With which method was the substrate mapping done.

The paper hypothesizes that using a heavier inert gas molecule mixed with the carrier gas will enhance the density and step coverage.

- Regarding the density:

How do the films compare with other amorphous boron carbide films?

To which degree would the author expect a density enhancement, considering that their films are already 90 % as dense as the bulk crystalline phase?

Note that for the same composition reference 28 gives a density of 2.51 g/cm³

- Regarding step coverage:

The authors show a beautiful result that clearly correlates with the addition of Xe. However, adding an inert gas to hydrogen – even at low concentrations – also implies that all gas phase properties are affected (e.g., thermal conductivity, viscosity), and their experiments do not seem to try to separate and exclude these effects. I would recommend the author to either emphasize the results and use the discussion as an attempt to explain the results OR consider all the alternative explanation and design experiments to attempt to falsify them.

- Using an exponent of 0.5 (square root) for Grahams law, the ratio of diffusion rates for Xe/TEB is 1.16 but TEB/Ar is even higher (1.56). Based on the author hypothesis, since Ar should diffuse even faster than TEB shouldn't the step coverage be much worse than without it?

Additionally, the authors observed improved uniformity and higher tensile stress:

- As for the improved step coverage the authors should discuss which property or combination of properties that could lead to this result, not only the change in diffusivity.
- An explanation for the significant increase in stress would be appreciated.

On Grahams law:

- It would be good to check the hypothesis under a more modern formalism, such as Chapman-Enskog
- Exponent deviates from 0.5 (square root) for lighter gas like H₂.

The authors suggest that 1 at. % is due to a change in chemistry. How many samples were measured? An alternative explanation is that ERDA, while being well-suited to measure lighter elements like boron, presents an added difficulty when it comes to boron carbides, which is the close proximity of the 11B and 12C signals, which makes the extraction of the data difficult and may slightly affect its accuracy. Was any measure taken to avoid this overlap?

(end)

Reviewer #3

(Remarks to the Author)

This study introduced a route for improving film uniformity and step coverage on structures characterized by large aspect ratios (AR). The underlying principle is straightforward, leveraging the atomic weight of Xe to enhance the diffusion of the reactive gas species. This method demonstrated efficacy in the author's first experiment subject with a 10:1 aspect ratio trench pattern. However, subsequent application of this deposition method to lateral high-aspect ratio structures (Pillar-Hall32 chips from Chipmetrics) yielded unclear results, characterized by insufficient detailed data. The assertion regarding conformal growth being achievable in larger AR than 50:1 lacks adequate data support, yet it represents a pivotal aspect capable of demonstrating a significant technological breakthrough. Furthermore, the comparative effectiveness of the method proposed by the authors in contrast to other strategies capable of achieving conformal films or even superconformal films remains unclear. Due to the above reason, please reconsider the novelty and noteworthy contribution of this study. Unfortunately, the paper cannot be accepted in its present form.

- 1、 The clarity of the introduction's logic requires enhancement, necessitating a comprehensive elucidation of the technical bottlenecks, the extent to which previous studies have addressed them, the identified shortcomings, and the potential advantages offered by this study.
- 2、 The author proposes that the heavier Xe atoms promote the diffusion of the TEB molecules and their gas phase decomposition products down the trench. What is the principle behind this? Is it adsorption that causes the reactive species to be carried and diffused by heavy species? Is there a scientific foundation supporting the schematic depicted in Fig.1, and why isn't it that most of the heavy species diffuse into the trench while the light species remain on the surface? Does this have anything to do with the amount of Xe additive?
- 3、 On page 4, the author proposes an alternative way to view the results in Fig.2, but lacks empirical support. Please redesign the experiment arrangement and provide an in-depth analysis of the principles behind the phenomenon.
- 4、 The author mentioned that 450 °C is the upper temperature limit for SC=1 in the previous study, which results in the insufficient density of the boron carbide films. Why did the author choose 550 °C in this study? If the temperature is further increased, will a higher-density film be deposited?
- 5、 The observed significant increase in tensile stress in the film following the addition of Xe is concerning as it may adversely impact the film's quality. Please meticulously analyze the underlying causes of this phenomenon to better understand the implications and potential mitigation strategies.
- 6、 The experiment results of the lateral high-aspect ratio structures (Pillar-Hall32 chips from Chipmetrics) should be reorganized. What exactly does Fig.4 a, b, and c illustrate? Why is it possible to conclude that the deposition would be conformal in a 50:1 aspect ratio feature? Is the film deposition also performed on the structures with AR=50:1? Detailed parameters of the film deposited on structures with AR=20:1 should be given.
- 7、 The characterization methods employed in this study are rather conventional, and certain results from these methods lack substantial meaningful information. For instance, The XPS outcomes depicted in Fig. 5 appear boring, and the corresponding discussion lacks a clear focus and fails to draw significant conclusions. Please reconsider the necessity of including these aspects in the analysis.

Reviewer #4

(Remarks to the Author)

Dear Authors,

Overall, the manuscript demonstrates interesting data and results, and the supplemental simulations give key insights to presented work but could incorporate a few changes. I have the following comments.

1. Authors should strongly consider streamlining content into sections namely abstract, introduction, methods, results, discussion, and conclusion.
2. Line 73 – Authors could potentially explain why SC decreases with increasing temperature as diffusion increases with higher temperature, suggesting better step coverage in features.
3. Line 78 – Authors might consider repeating diffusion modeling for no Xe, 100 sccm Argon and compare results with the diffusion modeling with 100 sccm Xe. This comparison along with step coverage from data sets could help provide better

insights to readers.

4.Methods – Authors could include a schematic/diagram for the CVD reactor used for ease of understanding of readers.

5.Line 118 – Authors could include uniformity maps for experiments with 100 sccm Argon dilution to provide concrete evidence of uniformity improvement via lateral dilution.

6.Line 125 – Authors might want to consider roughness improvement more critically as improvement in roughness ~ 0.7 nm might be within variance of measurement.

7.Figure 3 – Authors might want to highlight structure placement with and without dilution of Xe/Ar as thickness of deposition is not uniform in all cases, which might lead to comparisons and different deposition rates.

8.For future work authors might want to consider similar experiments at different temperatures, to understand if the effect of competitive diffusion is magnified/diminished at different temperatures.

Version 1:

Reviewer comments:

Reviewer #1

(Remarks to the Author)

The authors describe a way of increasing the step coverage of a CVD process by adding Xe to the gas mix. They claim the interaction between Xe and precursor molecules is responsible for this effect.

I enjoyed reading the updated version and would like to thank the authors for this work. This is a very exciting idea.

However, the presented experimental evidence is not yet fully convincing me.

Fig. 4

I appreciate the changes the authors made to this figure. Comments on the provided data:

The EDS provided has very high signal to noise. It may be a minor oversight of the authors not to include the EDS voltage to the methods, but an accelerating voltage of 3kV is not sufficient to get good EDS signal. This may explain the high S/N of that measurement. In case higher voltages were used for the EDS, please update the methods section. If 3kV was used, this measurement should be redone with a higher voltage (at least 15 kV).

Also, while technically a cross-section of an LHAR structure was added, the image provided does not answer the issue I was raising because of its lower magnification. Zoomed in cross-sections of the LHAR structure on the other hand are ideally suited for that (as in figure 4, first version). These are samples that are available already. Adding similar cross-sectional SEM images of the 1:50 LHAR structure would tremendously benefit the credibility of this work.

It is also not clear what the yellow color in 4b means.

For fig. 5: For the sake of transparency, I strongly suggest adding counts to the y-axis.

While XPS count rate in itself would not be meaningful, relative ratios would be.

The noise levels indicate that these curves have been scaled which is a bizarre choice given the point the authors want to make.

Other, earlier comments that need to be addressed:

"some experiments were conducted by delivering Xe in a pulsed manner". Which ones? Are these results displayed in the main text? Recommendation to clearly indicate which results had the Xe pulsed.

Reviewer #2

(Remarks to the Author)

The authors have responded to many of the reviewer concerns and clarified the manuscript. The experimental results are exciting, unambiguous, and very valuable to the field. The readership deserves to see these results.

However, this manuscript continues to lack a credible physical or chemical explanation for the observations. The authors continue to refer to a diffusion "enhancement" but adding a scattering center to a diffusion problem, here Xe, does not enhance gas phase diffusion. Also, the authors continue to refer to the effect of differential diffusion during a hypothetical start of the process. While the latter is true for a time scale of microseconds, the growth takes place over minutes, so the microsecond phenomenon is not relevant. Also, it is not possible to fill a chamber like this instantaneously, so the hypothetical start will not be the case in practice.

==> On this basis my recommendation is publish with required minor revision.

In an earlier review I raised some other mechanistic possibilities, such as changes in the mix of gas-phase reactive species, which does appear to be the case based on the more uniform deposition of film across the planar substrate.

For example: On a more subtle level, impact of a massive species with a surface is known to enhance desorption of adsorbates via the high-energy tail on the Maxwell-Boltzmann velocity distribution. If reactive intermediates have a long surface residence time due to strong adsorption, then their diffusivity in a trench will be lowered (in diffusion theory, one adds the mean residence time to the mean flight time between collisions). If Xe helps even modestly to lower the residence time, then the effective diffusivity in a trench will be increased. If the authors used a trench whose width is smaller than the gas-phase mean free path, then very few gas phase collisions within the trench would occur. The experimental test is whether the step coverage would improve in the presence of Xe collisions with the walls but the absence of any gas phase collisions with Xe.

My suggestion to the authors is to eliminate from the manuscript a mechanistic hypothesis, and instead to state that the mechanism is not understood at present, but possibilities worth considering are XYZ. From my perspective, that would be much more credible than to insist on a mechanism that is not physically supported.

Page Line numbers

S 5-6 The partial pressure ratios at the start of gas flow (a time scale of ~ microseconds) are not relevant to the problem at hand: trench coating takes place over minutes.

The authors state that the gradient in pressure is what leads to steady-state mass transport. That is very elementary (fundamental) and adds nothing to the discussion. There is no need to point out a "101" concept.

S 7 The experiments with pulsed gas flow are unable to test the possible role of competitive co-diffusion because the rate at which the chamber pressure can be changed is relatively slow. The authors do not state the chamber volume or the pumping speed, but typically the time constant for filling or evacuating a chamber is a fraction of a second, i.e., a million times greater than the time scale that is relevant in Figs. S4 and S5.

2 48 "tunning" should be "tuning"

3 67-68 "A molecule with higher mass, that is inert to the deposition chemistry, could be used as diffusion enhancement agent..." ==> The fundamental issue remains: adding a scattering species to the system does not increase diffusivity. The fact that transient differences exist on the ~ 1 microsecond time scale, as described in the SI, are not relevant because film growth takes place on the time scale of minutes. Thus, the fundamental assumption of the manuscript is not supported.

7 141 "form" should be "from"

7 158-9 You state "The changes in growth rate distribution over the large area is attributed to the enhanced diffusion on a reactor scale, rather than a kinetics effect." This is not supported by any theory that can be used to calculate the diffusion coefficient. Also, it does not consider the effect of gas plug flow. Generally in this type of reactor the plug flow transport is a large contribution for length scales of cm.

9 194 You state "This phenomenon is likely attributable to the inhibited back diffusion of precursor..." but this is not a sufficient statement to explain the results. The addition of Xe will reduce mobility in both "forward" and "back" directions, and transport calculations in steady state must use this lower value.

(end)

Reviewer #3

(Remarks to the Author)

The paper has been well revised according to my previous suggestion, which can be accepted in the present form.

Reviewer #4

(Remarks to the Author)

Dear Authors,

I am satisfied with the responses to the questions and issues raised in the initial review. The revised manuscript is easier to follow based on feedback from the reviewers.

Version 2:

Reviewer comments:

Reviewer #1

(Remarks to the Author)

I want to thank the authors for their efforts in accommodating my requests.

This manuscript describes exciting work. I have a couple minor comments that I would like to see changed, but no need to further discuss unless the authors wish to.

p3. from these consideration*s*

p10: though a deeper understanding require*s* further studies

p11: ... Although the B 1s and C 1s core level spectral intensity decreases inside the structure due to the higher XPS yield of Si ...

I assume this is because of the thinner film, and not so much because of the Si.

p10 : Consequently, it is possible to infer that conformal growth might be attainable even in structures with aspect ratios surpassing 50:1.

and p 11: ... to conformal film in at least a 50:1 aspect ratio feature

I believe the comparison with the non-enhanced case should be made in both cases. (as is done for the 1:10 AR structures, "from 0.71 to 0.97").

Reviewer #2

(Remarks to the Author)

The authors have modified, or in some cases deleted, many unfounded statements in response to my earlier review. Unfortunately, they continue to refer to diffusion "enhancement" as a possible mechanism even though a physical calculation would show a reduction in diffusivity upon the addition of a more massive gas. The revised manuscript refers to the possible role of several mechanisms but gives no estimates or measurements that would constitute a plausibility case for one or more mechanisms.

It is OK with me if the journal chooses to publish this work based on the strength of the unique and interesting experimental observations, which are well documented.

I am concerned, however, that the inclusion of statements which are not supported by physics will detract from the credibility of this work in the judgment of many readers. The seriousness of that issue should be judged by the journal and by the authors.

Version 3:

Reviewer comments:

Reviewer #2

(Remarks to the Author)

In this revised manuscript a few words have been substituted, but the essential problem that I described in earlier reviews remains fully articulated: the authors state that "we seek to promote the diffusion of precursor molecules down the recessed features to allow for conformal deposition..." But the addition of heavy molecules does not promote diffusion, it retards it.

Having given the authors a final opportunity to restructure their arguments, but seeing in response no such restructuring, I leave to the discretion of the editors what decision to make with respect to this work. I no longer wish to be consulted about it.

To restate my earlier review, the experimental results remain very interesting and worthy of publication, but the counter-factual nature of the explanation given by the authors seriously detracts from the credibility of the work.

Version 4:

Reviewer comments:

Reviewer #3

(Remarks to the Author)

no more comments.

RESPONSE TO REVIEWERS' COMMENTS

We extend our heartfelt gratitude to all the reviewers who dedicated their time and expertise to evaluate our work. Your thoughtful feedback and insightful comments have been instrumental in refining our manuscripts. As researchers, we recognize the immense value of constructive criticism, and we wholeheartedly appreciate the effort you've invested in helping us enhance the quality of our work. Each suggestion, every observation, and every recommendation has contributed to a more robust and impactful body of work. In this viewpoint, we present our revised findings, incorporating the insights gained from your assessments.

Answers to reviewer 1

Major comments:

A cross-section of the LHAR structure shown in fig. 4 a/b/c is missing but should be added.

We have reworked Figure 4. The revised figure now accurately represents the LHAR structure. The text discussing Fig. 4 has been slightly revised to reflect this.

I also strongly suggest an additional comparison of the obtained profiles/cross-sections with another LHAR substrate, but without Xe. That way the merits of the Xe addition can be fully appreciated.

This has been added to the revised manuscript.

In the supporting information, a diffusion model is developed and used to simulate the partial pressure. The authors state that surface reactions are ignored in this model. Is there any knowledge of TEB surface reaction coefficients that warrants such simplification? e.g. in the LHAR structure of 100 micron (AR 200) this simplification probably does not hold?

Yes, our diffusion model excludes surface reactions. While this simplification may seem drastic, it is grounded in our understanding of the precursor behavior. Triethylboron (TEB), like many chemical vapor deposition (CVD) precursors, exhibits a low sticking coefficient. Our experience with CVD using TEB strengthens this understanding. In other words, it has a very low tendency to adhere to the surface upon collision. In contrast, atomic layer deposition (ALD) precursors are

often designed with sticking probabilities closer to 1. These highly reactive species readily react and form stable bonds on the substrate surface.

Rationalizing the Assumption:

Given TEB's behavior, we make the assumption that its sticking probability is significantly less than 1. Consequently, the precursor molecules predominantly diffuse back in the gas phase rather than rapidly reacting at the surface. This justifies our diffusion model which does not consider surface reactions.

Implications for Calculations:

By proceeding with this assumption, we maintain computational tractability while capturing the dominant transport behavior. Our calculations provide valuable insights into precursor distribution, penetration depth, and film growth.

We have added a note on this in the revised supporting information.

It seems the H₂ partial pressure influences film composition through ligand elimination. What is the effect of the Xe co-flow on the H₂ diffusion? Authors should distinguish on the effect of Xe on both TEB and H₂ in their discussion. I feel this distinction is somewhat lacking now, both in the main text and supporting info.

The presence of H₂ affects film composition by influencing precursor dissociation and ligand elimination, as TEB can undergo both beta hydride elimination, without involvement of H₂, and hydrogen-assisted ligand removal, where H₂ is a reactant. Our previous studies at 450 °C revealed a 3:1 ratio between these reactions to occur. However, we strategically maintained two orders of magnitude higher flow rate for hydrogen gas compared to Xe and TEB to mitigate the hydrogen partial pressure's impact on the reaction rates across the trench depth. Additionally, hydrogen diffuses much faster than other gas species given its very low molecular mass. Importantly, we have not observed any evidence indicating that variations in hydrogen partial pressure across the trench structure directly cause surface reactions. Instead, it appears that the system becomes saturated due to excess H₂ pressure.

We have added a note on this in the revised supporting information.

Minor comments and suggestions:

frame b of figure 1 visually suggests that adding Xe leads to a higher pressure, but from the text I understood that the pressure remains the same. Soft suggestion to maybe reconsider this figure, so it includes H₂?

We have modified the figure to ensure that it does not give the impression of increased pressure when Xe is added. However, all our attempts to add H₂ to the figure made it less clear. Instead, we revised the figure caption to avoid the possible confusion that the reviewer points out.

Answers to reviewer #2

This manuscript reports an interesting and unobvious result: in the deposition of boron carbide films by thermal CVD, the coating profile within a deep recessed structure (a trench) can be made more conformal by adding an inert gas (Xe) to the process gases. The experimental data are credible, but the generality of the result and how it may vary over the large area of the substrate are not reported.

Our experiments this far have focused on understanding the mechanisms that deposit films with the desired property, we agree that a comprehensive study would involve understanding this effect over a large area. Here we are mainly limited by the size of our CVD reactor which has a maximal substrate area of about 3×10 cm. While our experimental data clearly demonstrate the efficacy of Xe addition, we suggest from Fig. 3, that the process will not only function, but even be beneficial over large areas. We have highlighted this more in the revised manuscript, around the discussion on Fig. 3.

Regarding the generality of these results, it has already been reported by researchers that the approach to controlling the kinetics of film growth lies in difference in diffusivity of the reactants.[1] By further extending that to the use of heavy inert gas as a diffusion additive it is reasonable to foresee a generality. We have included additional explanatory text to address this point towards the end of the discussion.

- In a microtrench, molecular collisions are typically with the wall (molecular flow conditions), not with other molecules in the gas phase. That can be estimated by simple mean free path calculations, which the authors have not done. Under molecular flow, the scattering effect of an inert gas is normally small and can be neglected.

Calculations:

$$\text{Mean free path } (\lambda) = \frac{RT}{\sqrt{2} \pi d^2 N_A P} = 6.612 \text{ } \mu\text{m}$$

Where, P = 5 kPa, T = 823 K (550 °C), d = 2.89 x 10⁻¹⁰, N_A = 6.022 × 10²³/mol

$$\text{Knudsen number (Kn)} = \frac{\lambda}{L} = 1 \text{ to } 1.1$$

Where, λ = 6.612 μm and the representative physical length scale (L) = 6 to 6.5 μm

The corresponding empirical flow regime classification is transitional flow (0.1 < Kn < 10).

The transitional flow allows for a more controlled process, as it avoids the unpredictability of turbulent flow while still being more dynamic than laminar flow. This regime finds a balance between effective mass transport and desired gas phase collisions.

In our investigation, we deliberately maintained a low mean free path for gas molecules, achieving mean free paths in the order of micrometers. Notably, the presence of excess hydrogen served a dual purpose. It minimized the mean free path to the micrometer scale and simultaneously nullified the hydrogen partial pressure dependency of deposition rate.

Given that our primary focus was on studying conformality enhancement through competitive diffusion, the micrometer-scale mean free path allowed us to operate the process at a Knudsen number of around 1. The resulting Knudsen number, approximately 1, indicates a transition regime where both intermolecular collisions and collisions with trench walls play significant roles. By meticulously controlling the mean free path, we aimed to create conditions conducive to studying gas-phase diffusion within confined microstructures. Remarkably, even without any diffusion additive, our process achieved a step coverage of 0.76 in a 10:1 aspect ratio trench structure, surpassing typical chemical vapor deposition (CVD) processes. This improvement was attributed to enhanced gas phase diffusion resulting from reduced sticking probability.^[2] Again, operating the CVD process at temperatures below the flux-limited reaction regime allowed molecules to rebound from the wall and engage in further collisions with both gas-phase molecules and the trench wall.

We have included additional explanatory details in the manuscript, covering flow conditions and the mean free path calculations performed.

-Outside of the trenches, the overall reaction kinetics are clearly varying upon addition of Xe, as evidenced by the significant change in growth rate over the large area of the substrate (Fig. 3a, "with Xe"). The reasons for this are not discussed.

Given that CVD precursor must diffuse through the boundary layer to reach the substrate surface, we argue the phenomenon of diffusion enhancement applies here as well. We suggest that the changes in growth rate distribution over the large area is due to enhanced diffusion on a global (reactor) scale, rather than a kinetics effect. By strategically maintaining a sufficiently low mean free path (achieved through excess hydrogen flow), we aimed to create a condition where precursor molecules undergo molecular collisions before it reaches the substrate surface which can be manipulated with a diffusion additive.

We have included additional text to explain and address this feedback. While our focus remains on conformality enhancement, we appreciate the broader implications of these phenomena.

The experimental details state that the plug-flow system is run at constant total pressure as controlled by a throttle valve with a feedback system. In this case, the pumping speed and therefore the partial pressure for each gas will vary with process conditions in a way that is not known unless non-trivial calibrations are performed.

Our process was conducted in a hydrogen background. The hydrogen flow rate was a few orders of magnitude higher than that of TEB and Xe. Consequently, the influence of plug flows other than hydrogen on partial pressure was negligibly low. To narrow down the focus around the implications of Xe addition, we maintained the total reactor pressure for all experiments. This approach allowed us to operate the processes at the same Knudsen number, facilitating meaningful comparison. We have added additional text to the manuscript to avoid this confusion.

I appreciate that it is not realistic to attempt to determine "all" the details of the reactive species mix. But there should be some discussion based on correct physics and chemistry, and preferably some parametric experiments, to sort out what is most important in this effect.

We are not certain that we fully understand what the reviewer means by this comment.

We have used excess H₂ partial pressure, resulting in no dependency between deposition rate and hydrogen partial pressure. While the concept of using a heavy molecule as diffusion enhancement was considered, we strategically opted for an inert gas (Xe). From these considerations, we narrowed down the variables that significantly contribute to the deposition process and allowed us to focus on the essential aspect. The choice of Xe simplified the experiment setup, allowing us to explore the specific impact of precursor concentration gradient without introducing additional complexities.

Our experimental observations demonstrate that the addition of Xe reduces the deposition at the trench top. We suggest that this reduction deposition rate occurs due to the change in ambient condition from TEB in H₂ to TEB in a Xe-H₂ mixture, pushing the TEB molecules and their decomposition products away from the surface, down the trench. This would reduce the effective sticking coefficient of TEB.

In our additional experiment with 50 sccm Xe flow, we found that the results were between those observed at 0 and 100 sccm flow, rendering a SC of 0.87 for 50 sccm Xe flow.

- What is the actual TEB flow/pressure? How does it compare to the inert gas flow and is H₂ in excess (the authors pointed out that TEB partially reacts with H₂)

TEB was introduced into the reactor at a rate of 1 sccm, resulting in a partial pressure of approximately 1.2 Pa. The Xe flow rate was 100 sccm, corresponding to a partial pressure of approximately 18 Pa. The total pressure was kept at 5 kPa for all depositions with a 2000 sccm flow (4980 Pa) regulated by a throttle valve on the process pump.

As you correctly pointed out, TEB reacts with Hydrogen. The excess flow rate prevents Hydrogen partial pressure dependency on film deposition rate. Since the pumping speed was throttled and ultimately partial pressure represents the gas concentration accurately, we have mentioned all the gas concentrations in terms of partial pressure in the methods session.

Growth rate: there is no absolute thickness mentioned in the text for the data on Fig.3, and the deposition time is not mentioned either.

The growth comparison with Xe, without Xe and Xe replaced by Ar are already presented in Figure 2, with actual thickness values and deposition time was mentioned in the figure caption as 60 min. Since Figure 3 compares the area of uniform film deposition, the thicknesses were

normalized to facilitate an easy comparison. In response to the comment, we have now added the actual growth rates in the paragraph.

- *With which method was the substrate mapping done.*

Although we mentioned this in the methods session, we did not elaborate. The thickness mapping was accomplished by verifying it through cross-section SEM measurements. The sample was broken into multiple pieces to perform cross sectional thickness measurements with 1 cm intervals. Three measurements (left edge, center, and right edge) were performed at each cross section. We have added more text to the methods session to explain this.

The paper hypothesizes that using a heavier inert gas molecule mixed with the carrier gas will enhance the density and step coverage.

- Regarding the density:

How do the films compare with other amorphous boron carbide films? To which degree would the author expect a density enhancement, considering that their films are already 90 % as dense as the bulk crystalline phase?

Allow us to clarify our stance on this matter.

While the addition of Xe as a heavier inert gas has been hypothesized to enhance film step coverage, we do not hypothesize that Xe addition enhances the density. Instead, we demonstrate that by using a diffusion additive we can achieve improved conformity without compromising film density.

In our previous investigations, we observed that operating the process at lower temperatures enhances film conformality. However, this came at the cost of reduced film density (see figures below showing results from earlier studies).^[2] Our new approach introduces the use of Xe as a diffusion additive to achieve conformal film deposition also at higher temperature to allow maintaining a higher film density. By introducing Xe at optimum partial pressure, we strike a balance between conformality and density, resulting in film that shows both desired properties. We hope this clarification addresses your concerns.

Figure showing film conformality as a function of deposition temperature.

Figure showing film density as a function of deposition temperature.

- Regarding step coverage:

The authors show a beautiful result that clearly correlates with the addition of Xe. However, adding an inert gas to hydrogen – even at low concentrations – also implies that all gas phase properties are affected (e.g., thermal conductivity, viscosity), and their experiments do not seem to try to separate and exclude these effects. I would recommend the author to either emphasize the results and use the discussion as an attempt to explain the results OR consider all the alternative explanation and design experiments to attempt to falsify them.

We have updated the discussions with discussions on the properties of the gas mixture.

- Using an exponent of 0.5 (square root) for Grahams law, the ratio of diffusion rates for Xe/TEB is 1.16 but TEB/Ar is even higher (1.56). Based on the author hypothesis, since Ar

should diffuse even faster than TEB shouldn't the step coverage be much worse than without it?

There is an additional factor to consider here: partial pressure.

TEB's partial pressure is two orders of magnitude lower than that of Ar. Statistically, this scarcity of TEB molecules prevents substantial enhancement of Ar diffusion. However, in the case of Xe and TEB, the dynamics are reversed, leading to the observed effects due to an excess of Xe pushing the majority of the TEB molecules. In this case, TEB molecules have a higher probability of colliding with heavier Xe than with TEB itself. On the other hand, the probability of Ar undergoing collisions with TEB is much less than that of collisions with Ar itself. This is due to the two orders of magnitude higher partial pressure for Ar compared to TEB. It is experimentally verified in the result presented in Fig. 2.

In addition, as we discussed in the manuscript, while the TEB has a higher mass than Ar, most of the gas phase reaction products of TEB are either comparable or lighter than Ar.

β -hydride elimination pathway^[3]

Hydrogen-assisted ethane elimination pathway^[3]

Since it is only a little decrease in the step coverage with Ar from the process without any diffusion additive due to the other underlying factors that we discussed above, we did not emphasize it in the manuscript in a way the reviewer kindly pointed out. The strategy is to deliver the lighter species in limited amount and the heavier species in excess.

Similar approach has been reported by W. B. Wang et al., see Ref^[1]. The impact of varying the molar mass of the of reactant on penetration depth of film deposition in ALD has been reported

by J. Yim et al.^[4] These studies were also considered the phenomena that lighter molecules diffuse faster than heavier ones.

***Additionally, the authors observed improved uniformity and higher tensile stress:
- As for the improved step coverage the authors should discuss which property or combination of properties that could lead to this result, not only the change in diffusivity.
- An explanation for the significant increase in stress would be appreciated.***

We are confused by this comment. To us it is obvious that improved diffusivity leads to better uniformity over a large area. Enhanced diffusion over larger areas is typically the key design aspect when designing large area CVD reactors.

The relative increase in stress is high, but the absolute level of stress is still very low, especially for boron carbide films which are notoriously known for their compressive stress. Generally, a higher growth rate over an extended area leads to an increased number of molecules approaching the surface, resulting in diminished adatom mobility. The relative increase in the film stress can be attributed to the decreased adatom mobility, which is due to the observed higher growth rate over a large area following the addition of Xe.

We have added this reasoning to the revised manuscript.

On Grahams law:

- It would be good to check the hypothesis under a more modern formalism, such as Chapman-Enskog

Indeed, we have considered the Chapman-Enskog formalism in our theoretical discussions provided in the supplementary information. While our current study does not delve into detailed mathematical modeling, the principles from Chapman theory inform our qualitative understanding.

Numerous works have been reported on the study of penetration depth or conformality of film deposition in the context where lighter molecules diffuse faster than heavier ones. Few examples are: theoretical work by J. Yim et al.^[4] and experimental studies by W. B. Wang et al.^[1]

While it is not realistic to quantify every aspect of even a simple system like this, the attempt under set of assumption aligns with experimental evidence and theoretical insights.

- Exponent deviates from 0.5 (square root) for lighter gas like H₂.

Despite this deviation, Graham's experimental observations remain valid. The complexity of CVD reactions, considering all gas-phase reaction byproducts, makes it challenging to quantitatively model the entire system. However, we can adopt a pragmatic approach. By considering the worst-case scenario – focusing on TEB and ignoring lighter gas-phase reaction products, we can still characterize the diffusion behavior. This simplification allows us to gain insight into the fundamental transport mechanisms. While real-world condition introduces additional complexities, the lighter TEB decomposition product turns the situation more favorable, and it was reflected in the experimental verification. This simplified analysis provides a very useful starting point.

In addition, although Graham's law was discussed, it was further supported with Chapman and Fick equations. Fick equation can be flexibly used in CVD flow regime especially in Knudsen flow regime. Similar observations have been reported by M. Ylilammi et al. ^[5] and J. Yim et al. ^[4]

The authors suggest that 1 at. % is due to a change in chemistry. How many samples were measured? An alternative explanation is that ERDA, while being well-suited to measure lighter elements like boron, presents an added difficulty when it comes to boron carbides, which is the close proximity of the 11B and 12C signals, which makes the extraction of the data difficult and may slightly affect its accuracy. Was any measure taken to avoid this overlap?

The ERDA measurements were conducted meticulously, considering both chemistry and instrument limitation. We carefully selected measurement conditions where the B and C signal could be extracted separately. Our experience from previous work also guided us. In our prior publication, we observed that the boron-to-carbon ratio shifted from B₄C to B₃C when transitioning from purely Hydrogen to purely Argon ambient.^[2] So, we believe that this subtle difference of 1 at. % is indeed expected due to the change in the ambience. We have expanded our discussion on this in the revised manuscript.

Since we used natural boron precursor and employed ERDA, we were able to separate boron-10 and boron-11 isotopes. The boron and the carbon concentrations were also verified by making sure that the boron-10 to boron-11 ratio is conserved.

Answers to reviewer #3

1、 The clarity of the introduction's logic requires enhancement, necessitating a comprehensive elucidation of the technical bottlenecks, the extent to which previous studies have addressed them, the identified shortcomings, and the potential advantages offered by this study.

We have rewritten parts of the introduction and hope that the logic is clearer now.

2、 The author proposes that the heavier Xe atoms promote the diffusion of the TEB molecules and their gas phase decomposition products down the trench. What is the principle behind this? Is it adsorption that causes the reactive species to be carried and diffused by heavy species? Is there a scientific foundation supporting the schematic depicted in Fig. 1, and why isn't it that most of the heavy species diffuse into the trench while the light species remain on the surface? Does this have anything to do with the amount of Xe additive?

The heavier Xe has a lower diffusivity due to its greater mass. TEB and its even lighter gas phase decomposition products diffuse more rapidly. By introducing Xe, we create a gradient concentration over trench depth. At the surface, Xe dominates due to its slower diffusion. As we move down the trench, TEB becomes more prevalent. This gradient modifies the local partial pressure of reactive species. Also, the difference in local concentrations causes variation in surface adsorption. Too little Xe, and the gradient won't be significant. Too much, and the Xe might dominate, hindering TEB diffusion. Optimum Xe concentration ensures controlled transport.

In summary, we leverage the difference in diffusivity to tailor the partial pressure gradient of reactive gas species along the trench depth. The elegant interplay of heavier and lighter species orchestrates efficient film deposition.

3、 On page 4, the author proposes an alternative way to view the results in Fig. 2, but lacks empirical support. Please redesign the experiment arrangement and provide an in-depth analysis of the principles behind the phenomenon.

We agree that our proposed alternative viewpoint based on Langmuir adsorption, in the manuscript needs to be expanded.

When Xe or Ar was introduced, it caused local dilution of TEB for adsorption at available surface sites. The variation in the deposition rate can be attributed to this. This competitive diffusion leads to changes in the precursor's concentration, which is seen as a reduction in the effective sticking coefficient. By considering Langmuir adsorption, we bridge the gap between gas-phase diffusion and surface interaction. Similar observations can be found in literatures that study CVD or ALD processes, a few references are Ref [4], Ref^[1] and Ref.^[6].

4、 The author mentioned that 450 ° C is the upper temperature limit for SC=1 in the previous study, which results in the insufficient density of the boron carbide films. Why did the author choose 550 ° C in this study? If the temperature is further increased, will a higher-density film be deposited?

The decision to increase the deposition temperature to 550 °C was deliberate, and it indeed leads to a higher density-film. We have investigated film density as a function of deposition temperature in our previous study, see figure below ^[7]. While 450 °C renders a film deposition with density of 1.9 g/cm³, 550 °C deposition temperature gives a density value of 2.28 g/cm³, see figure below.^[7]

Figure showing film conformality as a function of deposition temperature.

Figure showing density as a function of deposition temperature.

5. The observed significant increase in tensile stress in the film following the addition of Xe is concerning as it may adversely impact the film's quality. Please meticulously analyze the underlying causes of this phenomenon to better understand the implications and potential mitigation strategies.

As we replied to reviewer 2, the relative increase in stress is high, but the absolute level of stress is still very low, especially for boron carbide films which are notoriously known for their compressive stress. Generally, a higher growth rate over an extended area leads to an increased number of molecules approaching the surface, resulting in diminished adatom mobility. The relative increase in the film stress can be attributed to the decreased adatom mobility, which is due to the observed higher growth rate over a large area following the addition of Xe.

6. The experiment results of the lateral high-aspect ratio structures (Pillar-Hall32 chips from Chipmetrics) should be reorganized. What exactly does Fig.4 a, b, and c illustrate? Why is it possible to conclude that the deposition would be conformal in a 50:1 aspect ratio feature? Is the film deposition also performed on the structures with AR=50:1? Detailed parameters of the film deposited on structures with AR=20:1 should be given.

The figure has been updated by including an additional profile, which illustrates the penetration depth in the absence of Xe. This addition highlights the role of Xe as a diffusion enhancer. The beauty of pillar-hall structures lies in their ability to facilitate process validation. There is no need to employ a structure with a 50:1 aspect ratio; instead, a pillar-hall structure enables us to gauge the process's efficacy by assessing the penetration depth within the structure itself. The penetration depth of the conformal film deposition with Xe is 25 μm corresponding to a 50:1 aspect ratio.

The detailed parameters for the film deposition is explained in the methods session. It was maintained the same for the 20:1 aspect ratio structure as well.

Answers to reviewer #4

1. Authors should strongly consider streamlining content into sections namely abstract, introduction, methods, results, discussion, and conclusion.

The structure of our manuscript has been streamlined in accordance with the Nature Communications template. The content is organized into the following sections: Abstract, introduction, results, discussion and methods.

2. Line 73 – Authors could potentially explain why SC decreases with increasing temperature as diffusion increases with higher temperature, suggesting better step coverage in features.

The key to depositing films with SC = 1 into recessed features, e.g., trenches, by CVD, is to control the reactivity of the deposition chemistry. The key parameter to do this by is the temperature, as almost all chemical reactions are slower at lower temperatures. So, the typical approach to SC = 1 CVD films is to limit the reaction temperature to a minimum, effectively choking the CVD chemistry, preventing the molecules from reacting and allowing them to diffuse further. This is well known in CVD and has been extensively reviewed in e.g. ref 19 of the manuscript.

3. Line 78 – Authors might consider repeating diffusion modeling for no Xe, 100 sccm Argon and compare results with the diffusion modeling with 100 sccm Xe. This comparison along with step coverage from data sets could help provide better insights to readers.

Based on your comment, we have now included the additional modeling without Xenon (Xe). This is consistent with the results presented in Figure 2. Furthermore, we acknowledge the importance of considering the gas phase decomposition products of TEB. These products are lighter and not considered in the modeling.

Modeling with Ar does not provide additional insights for the following reason: TEB's partial pressure is two orders of magnitude lower than that of Ar. Statistically, this scarcity of TEB molecules prevents substantial displacement of Ar. However, in the case of Xe and TEB, the dynamics are reversed, leading to the observed effects due to an excess of Xe pushing the majority of the TEB molecules. In this case, TEB molecules have a higher probability of colliding with heavier Xe than with TEB itself. Thus, modeling with Xe allows us to observe a clear difference in flux. On the other hand, the probability of Ar undergoing collisions with TEB is much less than that of collisions with Ar itself. This is due to the two orders of magnitude higher partial pressure for Ar compared to TEB. It is experimentally verified in the result presented in Fig. 2.

We have updated our manuscript accordingly to reflect these insights.

Figure: TEB flux as a function of trench depth for various time.

Methods – Authors could include a schematic/diagram for the CVD reactor used for ease of understanding of readers.

We chose not to do this as a schematic of a CVD reactor is not very informative.

5.Line 118 – Authors could include uniformity maps for experiments with 100 sccm Argon dilution to provide concrete evidence of uniformity improvement via lateral dilution.

The experiment with 100 sccm Argon mirrors that of the experiment conducted without Xe and shows no notable difference.

6.Line 125 – Authors might want to consider roughness improvement more critically as improvement in roughness ~ 0.7 nm might be within variance of measurement.

This difference in film stress can be correlated with the observed changes in surface roughness, suggesting that the improvement is not merely a result of measurement variance but is indicative of a genuine change in the film's properties.

7. Figure 3 – Authors might want to highlight structure placement with and without dilution of Xe/Ar as thickness of deposition is not uniform in all cases, which might lead to comparisons and different deposition rates.

In our methodology, we employ a dimensionless parameter known as step coverage, which is designed to account for the difference in the coating thickness.

Additionally, we would like to clarify that all structures were positioned identically within the reactor to minimize any discrepancies in deposition conditions. This consistent placement is critical for ensuring that any observed differences in deposition rates are attributable to the experimental variables under investigation rather than positional effects within the reactor.

8. For future work authors might want to consider similar experiments at different temperatures, to understand if the effect of competitive diffusion is magnified/diminished at different temperatures.

Very good idea!

[1] W. B. Wang, N. N. Chang, T. A. Coddling, G. S. Girolami, J. R. Abelson, *J. Vac. Sci. Technol. A Vacuum, Surfaces, Film.* **2014**, 32, 051512.

- [2] A. H. Choolakkal, H. Högberg, J. Birch, H. Pedersen, *J. Vac. Sci. Technol. A* **2023**, *41*, 013401.
- [3] M. Imam, K. Gaul, A. Stegmüller, C. Höglund, J. Jensen, L. Hultman, J. Birch, R. Tonner, H. Pedersen, **2015**, 1–22.
- [4] J. Yim, E. Verkama, J. A. Velasco, K. Arts, R. L. Puurunen, *Phys. Chem. Chem. Phys.* **2022**, *24*, 8645–8660.
- [5] M. Ylilampi, O. M. E. Ylivaara, R. L. Puurunen, *J. Appl. Phys.* **2018**, *123*, 205301.
- [6] A. Yanguas-Gil, Y. Yang, N. Kumar, J. R. Abelson, *J. Vac. Sci. Technol. A Vacuum, Surfaces, Film.* **2009**, *27*, 1235–1243.
- [7] A. H. Choolakkal, H. Högberg, J. Birch, H. Pedersen, *J. Vac. Sci. Technol. A* **2023**, *41*, 013401.

RESPONSE TO REVIEWERS' COMMENTS

We again extend our heartfelt gratitude to all the reviewers who dedicated their time and expertise to evaluate our work. Your thoughtful feedback and insightful comments have been instrumental in refining our manuscripts, and we are very happy to see that you share our enthusiasm for the results. As researchers, we recognize the immense value of constructive criticism, and we wholeheartedly appreciate the effort you've invested in helping us enhance the quality of our work. We have taken the advise for the reviewers and changed the manuscript accordingly, we hope that it will be found suitable for publication.

Additional author added

The revision required some additional measurements to be done. To reflect the work needed, and the additional discussions on the results, we added Samira Dorri as a new author to the revised manuscript. She was earlier mentioned in the acknowledgement, but we feel that her efforts during the second revision of the manuscript fully warrant co-authorship.

REVIEWERS COMMENTS AND OUR REPLIES

Reviewer #1:

The authors describe a way of increasing the step coverage of a CVD process by adding Xe to the gas mix. They claim the interaction between Xe and precursor molecules is responsible for this effect.

I enjoyed reading the updated version and would like to thank the authors for this work. This is a very exciting idea. However, the presented experimental evidence is not yet fully convincing me.

Fig. 4

I appreciate the changes the authors made to this figure. Comments on the provided data:

The EDS provided has very high signal to noise. It may be a minor oversight of the authors not to include the EDS voltage to the methods, but an accelerating voltage of 3kV is not sufficient to get good EDS signal. This may explain the high S/N of that measurement. In case higher voltages were used for the EDS, please update the methods section. if 3kV was used, this measurement should be redone with a higher voltage (at least 15 kV).

The accelerating voltage was 5 kV to limit the interaction volume, considering the thinness of the film. However, we recognized that this resulted in a higher signal to noise ratio.

In response to your recommendation, we have remeasured the sample using an accelerating voltage of 15 kV. We have updated figure 4 and the methods section to reflect this.

Also, while technically a cross-section of an LHAR structure was added, the image provided does not answer the issue I was raising because of its lower magnification. Zoomed in cross-sections of the LHAR structure on the other hand are ideally suited for that (as in figure 4, first version). These are samples that are available already. Adding similar cross-sectional SEM images of the 1:50 LHAR structure would tremendously benefit the credibility of this work.

The significant difference in the order of magnitude between the lateral dimension and the film thickness limited our earlier attempts. We have now reworked Fig. 4 and included the cross section in the best possible way.

It is also not clear what the yellow color in 4b means.

Regarding the yellow shade in Figure 4b., it is due to contrast enhancement. The boron carbide film deposition on the Si surface typically gives lesser contrast in gray scale. So, the gray scale micrographs were processed to duo-chrome images for better representation. We have added a sentence in the figure caption to clarify this.

For fig. 5: For the sake of transparency, I strongly suggest adding counts to the y-axis. While XPS count rate in itself would not be meaningful, relative ratios would be. The noise levels indicate that these curves have been scaled which is a bizarre choice given the point the authors want to make.

We have added counts to the y-axis in the revised Fig 5.

Other, earlier comments that need to be addressed:

"Some experiments were conducted by delivering Xe in a pulsed manner". Which ones? Are these results displayed in the main text? Recommendation to clearly indicate which results had the Xe pulsed.

The experiments conducted by delivering Xe in a pulsed manner are not included in the main text but are detailed in the supplementary information. It is now mentioned in the methods section.

Reviewer #2:

The authors have responded to many of the reviewer's concerns and clarified the manuscript. The experimental results are exciting, unambiguous, and very valuable to the field. The readership deserves to see these results.

However, this manuscript continues to lack a credible physical or chemical explanation for the observations. The authors continue to refer to a diffusion "enhancement" but adding a scattering center to a diffusion problem, here Xe, does not enhance gas phase diffusion. Also, the authors continue to refer to the effect of differential diffusion during a hypothetical start of the process. While the latter is true for a time scale of microseconds, the growth takes place over minutes, so the microsecond phenomenon is not relevant. Also, it is not possible to fill a chamber like this instantaneously, so the hypothetical start will not be the case in practice.

==> On this basis my recommendation is publish with required minor revision.

In an earlier review I raised some other mechanistic possibilities, such as changes in the mix of gas-phase reactive species, which does appear to be the case based on the more uniform deposition of film across the planar substrate.

For example: On a more subtle level, impact of a massive species with a surface is known to enhance desorption of adsorbates via the high-energy tail on the Maxwell-Boltzmann velocity distribution. If reactive intermediates have a long surface residence time due to strong adsorption, then their diffusivity in a trench will be lowered (in diffusion theory, one adds the mean residence time to the mean flight time between collisions). If Xe helps even modestly to lower the residence time, then the effective diffusivity in a trench will be increased. If the authors used a trench whose width is smaller than the gas-phase mean free path, then very few gas phase collisions within the trench would occur. The experimental test is whether the step coverage would improve in the presence of Xe collisions with the walls but the absence of any gas phase collisions with Xe.

My suggestion to the authors is to eliminate from the manuscript a mechanistic hypothesis, and instead to state that the mechanism is not understood at present, but possibilities worth considering are XYZ. From my perspective, that would be much more credible than to insist on a mechanism that is not physically supported.

We agree with your recommendations and appreciate the broader perspective they provide. CVD chemistry involves complex co-effects, and the overall pathway is likely a result of these collective effects. Several possible phenomenon and co-effects, which occurs simultaneously or

independently, warrant consideration. This includes the enhancement of adsorbates desorption rate and the reduction of surface residence time for reactive intermediates.

Your feedback inspired additional ideas, leading us to rewrite some portions of the manuscript, as per your recommendation, to better reflect these insights:

- We tried to soften our introduction of the hypothesis of diffusion enhancement in the introduction. Starting from Fick's law of diffusion is, in all honesty, how we got the idea to do the experiments in the first place. But the comments from reviewer 2 shows that we might have been too focused on this model to explain the results.
- We added the explanation model of reduced surface residence time as an alternative explanation in the results section and speculate that the full mechanism is likely a combination of these plausible explanations.
- We highlighted the need for more experiments and modelling in the Discussion section, to better reflect that our manuscript is only a start to understand and explore the effects we have noted.

Page Line numbers

S 5-6 The partial pressure ratios at the start of gas flow (a time scale of ~ microseconds) are not relevant to the problem at hand: trench coating takes place over minutes.

We agree with your comment. While trench coating takes several minutes, the shown effect is short lasting. On the other hand, our calculations, limited to a feasibility test of Xe as a co-diffusion pair for TEB, assume no surface reactions. However, in reality, steady-state mass transport due to film deposition and lighter reactive species generation occurs.

We have added additional text in supplementary information to clarify this.

The authors state that the gradient in pressure is what leads to steady-state mass transport. That is very elementary (fundamental) and adds nothing to the discussion. There is no need to point out a "101" concept.

We have now removed the statement regarding the gradient in pressure leading to steady state mass transport and have modified the paragraph portion accordingly.

S 7 The experiments with pulsed gas flow are unable to test the possible role of competitive co-diffusion because the rate at which the chamber pressure can be changed is relatively slow. The authors do not state the chamber volume or the pumping speed, but typically the time constant for filling or evacuating a chamber is a fraction of a second, i.e., a million times greater than the time scale that is relevant in Figs. S4 and S5.

The pulsed delivery basically demonstrates that we can regenerate the trenches from the saturation of Xe partial pressure by stopping the Xe supply for 30 seconds between two adjacent 1 second pulses.

Here the chamber pressure is not varying. It is maintained by the H₂ base pressure and a throttle valve. We pulse Xe gas instead of continuous supply. That will reflect in the Xe concentration in the chamber. Every time we pulse Xe, an increased transient concentration in the trench occurs alongside the rising edge of the Xe partial pressure spike in the reactor. On the negative side, we also note that this pulsed supply of Xe drastically decreased the deposition rate.

2 48 "tunning" should be "tuning"

It has been corrected in the revised manuscript.

3 67-68 "A molecule with higher mass, that is inert to the deposition chemistry, could be used as diffusion enhancement agent..." ==> The fundamental issue remains: adding a scattering species to the system does not increase diffusivity. The fact that transient differences exist on the ~ 1 microsecond time scale, as described in the SI, are not relevant because film growth takes place on the time scale of minutes. Thus, the fundamental assumption of the manuscript is not supported.

The calculations presented in our manuscript were performed to demonstrate the possibility of creating a concentration gradient as a function of trench depth. It is important to note that we explicitly stated that surface reactions were not taken into account in these calculations.

In a real-case scenario, film formation will occur, leading to a pressure drop at near surface due to the conversion of triethyl boron (TEB) into a solid film. This will enable more diffusive mass transport, which will follow a transient behavior with Xe. Meaning that it is not limited to a microsecond timescale in the real process. While we agree that this effect will not be as prominent as the transient differences observed during the initial few microseconds, it remains a significant factor during the diffusive steady-state mass transport.

Regardless, we now expanded the discussions not to fully rely on the mechanistic model. We now acknowledge that several possible phenomenon and co-effects, which occurs simultaneously or independently, warrant consideration. This includes the enhancement of adsorbates desorption rate and the reduction of surface residence time for reactive intermediates resulting in increased diffusion.

7 141 "form" should be "from"

We realize that both the authors and reviewer made an error. The correct term should be 'forms' instead of 'form'.

7 158-9 You state "The changes in growth rate distribution over the large area is attributed to the enhanced diffusion on a reactor scale, rather than a kinetics effect." This is not supported by any theory that can be used to calculate the diffusion coefficient. Also, it does not consider the effect of gas plug flow. Generally, in this type of reactor the plug flow transport is a large contribution for length scales of cm.

Considering your comments we have reworked the discussion. We now propose that the reduction in surface residence time for reactive intermediates, due to addition of heavy Xe gas, could lead to a higher lateral spread. This modified surface adsorption-desorption profile due to Xe addition may further facilitate more uniform film deposition. Alternatively, since solid-gas interface also influence the temperature uniformity in the reactor, the modified boundary layer induced by the more viscous Xe plug flow could potentially favor more uniform film deposition, even though this was not an intentional objective. These kinetic and thermodynamic factors collectively contribute to the observed phenomena.

9 194 You state "This phenomenon is likely attributable to the inhibited back diffusion of precursor..." but this is not a sufficient statement to explain the results. The addition of Xe will reduce mobility in both "forward" and "back" directions, and transport calculations in steady state must use this lower value.

We agree with your comment that the explanation of inhibited back diffusion is not a sufficient statement. We rephrased statements to better reflect the speculative nature.

Reviewer #3:

The paper has been well revised according to my previous suggestion, which can be accepted in the present form.

Thank you for your feedback!

Reviewer #4:

Dear Authors,

I am satisfied with the responses to the questions and issues raised in the initial review. The revised manuscript is easier to follow based on feedback from the reviewers.

Thank you for your feedback!

RESPONSE TO REVIEWERS' COMMENTS

We thank the reviewers for again generously giving of their time to provide feedback on our manuscript. Your efforts have helped us to improve our manuscript. Our responses to your latest comments are given below. We hope that our revised manuscript is now found suitable for publication.

Kind regards

Henrik Pedersen

Reviewer #1

I want to thank the authors for their efforts in accommodating my requests.

This manuscript describes exciting work. I have a couple minor comments that I would like to see changed, but no need to further discuss unless the authors wish to.

*p3. from these consideration*s**

Thank you for pointing out this error – it is fixed in the revised manuscript.

*p10: though a deeper understanding require*s* further studies*

Thank you for pointing out this error – it is fixed in the revised manuscript.

p11: ... Although the B 1s and C 1s core level spectral intensity decreases inside the structure due to the higher XPS yield of Si ...

I assume this is because of the thinner film, and not so much because of the Si.

Exactly – this has been clarified in the revised manuscript.

p10 : Consequently, it is possible to infer that conformal growth might be attainable even in structures with aspect ratios surpassing 50:1.

and p 11: ... to conformal film in at least a 50:1 aspect ratio feature

I believe the comparison with the non-enhanced case should be made in both cases. (as is done for the 1:10 AR structures, "from 0.71 to 0.97").

Exactly – this has been clarified in the revised manuscript.

Reviewer #2

The authors have modified, or in some cases deleted, many unfounded statements in response to my earlier review. Unfortunately, they continue to refer to diffusion "enhancement" as a possible mechanism even though a physical calculation would show a reduction in diffusivity upon the addition of a more massive gas. The revised manuscript refers to the possible role of several mechanisms but gives no estimates or measurements that would constitute a plausibility case for one or more mechanisms.

It is OK with me if the journal chooses to publish this work based on the strength of the unique and interesting experimental observations, which are well documented. I am concerned, however, that the inclusion of statements which are not supported by physics will detract from the credibility of this work in the judgment of many readers. The seriousness of that issue should be judged by the journal and by the authors.

We think that we better understand the point that the reviewer is trying to make now. Our use of diffusion enhancement was primarily an effort to describe what we observed from the experiments: higher film growth at the bottom of a trench without using any strategy to prevent growth in the trench openings seems, at least to us, to suggest that the diffusion of the precursors is enhanced. As we also wrote in the last version of our manuscript, we got the idea for the experiments from Graham's law of diffusion and thus we have considered our results from this viewpoint.

We agree with the reviewer that we have not shown any strict evidence for an enhanced diffusion, and we have therefore removed this formulation in the new revision of the manuscript.

RESPONSE TO REVIEWERS' COMMENTS

We thank the reviewers for again generously giving of their time to provide feedback on our manuscript. Your efforts have helped us to improve our manuscript. Our responses to your latest comments are given below.

We hope that our revised manuscript is now found suitable for publication.

Kind regards

Henrik Pedersen

Reviewer #2:

In this revised manuscript a few words have been substituted, but the essential problem that I described in earlier reviews remains fully articulated: the authors state that "we seek to promote the diffusion of precursor molecules down the recessed features to allow for conformal deposition..." But the addition of heavy molecules does not promote diffusion, it retards it.

Having given the authors a final opportunity to restructure their arguments, but seeing in response no such restructuring, I leave to the discretion of the editors what decision to make with respect to this work. I no longer wish to be consulted about it.

To restate my earlier review, the experimental results remain very interesting and worthy of publication, but the counter-factual nature of the explanation given by the authors seriously detracts from the credibility of the work.

We are very sorry that we were not able to revise our manuscript to meet your comments. We would like to think that this is largely due to the form of written communication via an editor that we have used. In a face to face direct communication we would hopefully have been able to see your points more clearly and modify arguments more in line with those points.

Reviewer #3:

I have carefully check the responses of the author and the suggestion of the reviewer, finding that there is really a critical question that the authors should answered:

The authors mentioned the Grahams law of diffusion, which indicates that the larger the molecular mass, the smaller the diffusion rate. This is in agreement with the reviewer's query about "the addition of heavy molecules does not promote diffusion, it retards it" "the physic", and have contradictions with the major conclusion of the present study. I also suggest that the authors should provide more explanations.

We agree with your analysis that Reviewer #2 seems to argue that we claim that addition of a heavy molecule promotes diffusion. But this is not how we mean. We argue that lighter molecules diffuse faster than heavier – from Graham's law of diffusion. Adding heavier molecules to the gas mixture thus disturbs the diffusion of the total gas mixture. From this we formed the hypothesis that led us to do the experiments. We argue that our results could be

explained by this disturbance. We cannot understand how this is not physically correct, as we see it as a consequence of Graham's law of diffusion.

Guided by the comments by Reviewer #2 we understood that we wrote our original manuscript far too stuck on this way of explaining our results (formulated hypothesis and confirmed hypothesis). Reviewer #2 presented the alternative explanation that the heavy Xe atoms can promote desorption of film growth species from the surface. We fully agree that this is a plausible explanation for the results and have incorporated it in the revisions. We now try to present both these explanations for the results, and that the mechanism is likely a mix of them. We cannot see which more explanations we should provide.